# Short Half-Life of Des-γ-Carboxy Prothrombin Is a Superior Factor for Early Prediction of Outcomes of Hepatocellular Carcinoma Treated with Radiofrequency Ablation

**DOI:** 10.3390/diagnostics13040696

**Published:** 2023-02-12

**Authors:** Chih-Chien Yao, Jing-Houng Wang, Chien-Hung Chen, Chao-Hung Hung, Yi-Hao Yen, Kwong-Ming Kee, Tsung-Hui Hu, Ming-Chao Tsai, Yuan-Hung Kuo, Sheng-Nan Lu

**Affiliations:** Division of Hepatogastroenterology, Department of Internal Medicine, Kaohsiung Chang Gung Memorial Hospital and Chang Gung University College of Medicine, Kaohsiung 833401, Taiwan

**Keywords:** hepatocellular carcinoma, radiofrequency ablation, treatment response, predictor, prothrombin produced by vitamin K absence or antagonism II, half-life

## Abstract

Background: The role of des-γ-carboxy prothrombin (DCP) in patients undergoing radiofrequency ablation (RFA) for hepatocellular carcinoma (HCC) needs to be clarified. Materials and methods: 174 HCC patients that underwent RFA were enrolled. We calculated the HLs of DCP from the available values before and on first day after ablation and assessed the correlation between HLs of DCP and RFA efficacy. Results: Of 174 patients, 63 with pre-ablation DCP concentrations of ≥80 mAU/mL were analyzed. The ROC analysis showed the optimal cut-off value of HLs of DCP for predicting RFA response was 47.5 h. Therefore, we defined short HLs of DCP < 48 h as a predictor of favorable treatment response. Of 43 patients with a complete radiological response, 34 (79.1%) had short HLs of DCP. In 36 patients with short HLs of DCP, 34 (94.4%) had a complete radiologic response. The sensitivity, specificity, accuracy, positive predictive value, and negative predictive value were 79.1%, 90.0%, 82.5%, 94.4%, and 66.7%. During the 12-month follow-up, patients who had short HLs of DCP had a better disease-free survival rate than patients with long HLs of DCP (*p* < 0.001). Conclusions: Short HLs of DCP < 48 h calculated on the first day post-RFA are a useful predictor for treatment response and recurrence-free survival after RFA.

## 1. Introduction

Hepatocellular carcinoma (HCC) is one of the most prevalent malignancies worldwide and the second most common cause of cancer-related death globally [1,2,3]. The current curative treatment modalities of early HCC include surgical resection and percutaneous ablation therapies, such as percutaneous ethanol injection (PEI), radiofrequency ablation (RFA), cryotherapy, and microwave coagulation [4,5]. Although hepatectomy achieves favorable outcomes for HCC, only approximately 40% of well-selected patients could be candidates for surgical resection [6,7]. RFA is a thermal treatment technique and it has widely been used to treat patients with small or unresectable HCC with curative intent [8,9]. However, the long-term survival of RFA is unsatisfactory because of the high incidence of tumor recurrence. Treatment failure of RFA, including local residual tumors or distant recurrence, is known to occur in 2.4–27% of patients after achieving complete percutaneous treatment initially [10,11]. Therefore, whether tumor markers could be a better alternative way of assessing treatment responses and early detection of recurrence persist as an important issue in the local treatment of HCC [5,12].

Alpha-fetoprotein (AFP) is the most commonly used tumor marker to diagnose HCC and it is also useful for prediction of survival and treatment response after local ablation [13,14,15]. However, AFP has suboptimal performance and some limitations. First, AFP has unsatisfactory sensitivity and specificity in clinical use [16,17]. Second, serum AFP concentration could be normal in patients with early-stage HCC [18]. Third, serum AFP also rises in patients with acute inflammation post-RFA, chronic liver disease and cirrhosis [19,20,21]. Des-gamma-carboxy prothrombin (DCP), also known as prothrombin produced by vitamin K absence or antagonism II (PIVKA II), which is produced by HCC, was demonstrated by Liebman et al. to be a useful marker for the diagnosis of HCC in 1984 [22,23]. DCP has some advantage over AFP in detecting HCC. First, DCP is more accurate than AFP in differentiating patients with HCC from those with nonmalignant chronic liver disease and those with other malignant tumors [24]. Second, DCP is an alternative marker used to detect newly developed or recurrent non-AFP-secreting HCC. Whether DCP is superior to AFP or not is still controversial, and measurement of both postoperative AFP or DCP has been reported as a useful predictor of survival and tumor recurrence [25,26]. However, the rate of tumor marker changes, which is more important than their absolute value, was reported in Johnson’s study because the degradation of serum tumor markers requires some time [27]. In our previous reports, we determined that a favorable AFP decrease is a good reference for monitoring the treatment effects of transcatheter arterial chemoembolization (TACE) and radiofrequency ablation (RFA) [28,29]. Shim et al. also reported that the half-life (HL) of AFP reflects prognosis after liver resection for HCC [30]. Furthermore, because of the shorter HL of DCP (2 to 4 days compared to AFP at approximately 5–7 days) [31], DCP has a potential advantage and may be superior to AFP in monitoring treatment responsiveness in time.

Currently, to our knowledge, studies on the HLs of DCP post-RFA and the correlation with the treatment response of HCC are still limited. Thus, the present study aims to assess the clinical significance of HLs of DCP post ablation and the association between the efficacy of treatment in a single center.

## 2. Materials and Methods

The study enrolled 174 patients with newly developed HCC or recurrent HCC who underwent ultrasound-guided percutaneous RFA as first-line curative therapy prospectively in Kaohsiung Chang Gung Memorial Hospital between September 2016 and January 2017. HCC diagnosis and staging were based on typical imaging pattern of HCC on dynamic computed tomography (CT) or enhanced magnetic resonance imaging (MRI), or histologic examination by needle biopsy. The criteria for RFA procedures were as follows: (i) single nodular HCC <5 cm in maximum diameter; (ii) multinodular HCC (<3 in number, each <3 cm in maximal diameter; (iii) noncirrhotic or Child-Pugh class A or B cirrhosis; and (iv) absence of portal vein thrombosis or distant metastasis. Patients who were administered warfarin and vitamin K therapy were excluded. This study was approved by the Ethics Committee of the Chang Gung Memorial Hospital, and informed consent was obtained from all patients. The baseline relevant variables included age, gender, serum laboratory tests consisting of aspirate amino-transferase (ALT), prothrombin time, total bilirubin, albumin, platelet (PLT) counts, AFP levels, HBV and HCV serology, cirrhosis, Child–Pugh classification, the Fibrosis-4 (FIB-4) Index, the ALBI(Albumin-Bilirubin) grade, tumor characteristics (including tumor size and number), and Barcelona Clinic Liver Cancer (BCLC) stage. Cirrhosis was diagnosed by ultrasound (US). A US scoring system consisting of liver surface, parenchymal, vascular structure, and spleen size was used to describe the severity of hepatic parenchymal damage [32].

### 2.1. Serum DCP Concentration Assay and Calculation of Half-Life

The serum DCP levels were measured by ARCHICTECT PIVKA-II immunoassay (ARCHITECT PIVKA-II; Abbott Laboratories, Chicago, IL, USA). We adopted the cut-off value of <40 mAU/mL for DCP as the normal range according to manufacturer’s instructions. Serum HL of DCP were calculated by using the following exponential and logarithmic formula, as previously described [30,33]: HL = 0.3 × 1/log10(C0/C1), where C0 is the baseline marker concentration (obtained less than 1 week before RFA), and C1 is the concentration on the first day post RFA. (The C1 interval was calculated from the time of complete RFA therapy to the time of the blood draw measurement at the first day post-ablation.) This formula is based on an exponential decay process: N(t) = N0 (1/2)^t/t1/2^, where N(t) is the amount at time t, N0 is the initial amount, and t1/2 is the HL of the substance measured [34].

### 2.2. RFA Techniques

RFA was performed on an inpatient basis using a Cool-tip ^TM^ RF Ablation System (Medtronic, Minneapolis, MN, USA), Big-tip (RF Medical Co., Seoul, Korea), or Viva RF electrode system (STARmed, Seoul, South Korea). Procedures were performed percutaneously under local anesthesia by qualified hepatologists with the guidance of real-time ultrasonography (US) using a 17-gauge, 2 or 3 cm needle. Temperature and tissue impedance were monitored during ablation. A tip temperature of 10–20 °C was maintained by infusing it in chilled water. Radiofrequency energy was delivered for 6–12 min for each application. For large lesions, multiple overlapping ablation zones were required.

### 2.3. Assessment of Treatment Efficacy at One Month Post RFA

Tumor response to RFA was assessed by standard imaging modality and a CT or MRI scan at 1 month after ablation [35]. A complete radiological response was defined as the complete absence of contrast enhancement of all known lesions on radiological grounds. We assessed the correlation between different HLs of DCP and treatment outcomes in terms of standard imaging modality.

### 2.4. Analysis of Long-Term Therapeutic Outcome at 12 Months Post RFA

We also evaluate long-term therapeutic response post-ablation for 12 months by recurrence-free survival (RFS). RFS was defined as time from date of RFA therapy to first HCC recurrence. Patients underwent follow-up ultrasound treatment every 3 months in the first year post RFA. If recurrent tumor was suspected, additional different imaging modality including CT or MRI was immediately performed to confirm the diagnosis. The tumor recurrence was classified into three different types: local tumor progression (LTP), intrahepatic distant recurrence (IDR) or extrahepatic recurrence (ER). LTR was defined as the reappearance of tumor along the margin of the ablation zone after RFA. IDR was defined as the occurrence of HCC in different liver segments from RFA ablation zones. ER was defined as metastatic tumor outside the liver. Recurrent HCC was treated by subsequent repeat RFA, surgical resection, transarterial chemoembolization (TACE), or systemic therapy (e.g., target or chemotherapy).

### 2.5. Statistical Analysis

Values were calculated as the mean ± standard deviation, proportion, or median (range). For intergroup comparisons, the chi-squared test and Fisher’s exact test were applied to analyze categorical variables, while Student’s *t*-test and the Mann–Whitney *U*-test were used for continuous variables with normal and skewed distributions, respectively. The recurrence-free survival curves were analyzed using the Kaplan–Meier method and compared using the log-rank test. A two-sided *p* value lower than 0.05 was considered to be significant. The statistical analysis was performed with SPSS or R version 2.14.1 for Windows (SPSS, Chicago, IL, USA).

## 3. Results

### 3.1. Patients Characteristics

Of these 174 patients, patients who had normal concentrations of DCP (n = 65, 65/174 = 37.4%) were excluded. The remaining 109 patients were included in further analysis (Figure 1). Of the 109 patients, 77 had well treatment effect with complete radiologic response (70.6%) at 1-month post-RFA therapy, and 32 had incomplete ablation (29.4%). During the long-term follow-up, 45 of 77 well-ablation patients (58.4%) had no recurrence at 12 months post-RFA therapy. The baseline characteristics of 109 HCC patients compared between groups with and without complete radiological response at 1 month post RFA by standard imaging modality are shown in Table 1. There were no significant differences between the groups among patient-related factors in terms of age, gender, underlying comorbidities (ex: obesity, hypertension, diabetes), virus-related hepatitis, percentage of cirrhosis, or blood biochemistry tests. However, patients without complete radiologic response had higher ALBI grade level were noted. (ALBI grade 1/2 were 75.3%/24.7% in complete response group and 50%/50% in incomplete response group, respectively, *p* = 0.012.) Among the tumor-related factors, there were no significant differences between groups in terms of treatment experience, tumor size, or BCLC staging. However, the rate of multiple lesions was higher in the incomplete response group (53.1% compared to 29.9%, *p* = 0.022). In addition, according to AJCC 8th edition staging system for HCC, the percentage of Stage II was significantly higher in the incomplete response group (46.9% compared to 22.0%, *p* = 0.018).

Although absolute levels of AFP and DCP at Pre-RFA and at 1 day post RFA displayed no significant difference between groups, the DCP value decrease in complete response group was more significant than that of the incomplete group (ΔDCP = 204.5 and 27.9 in complete and incomplete response group, respectively, *p* = 0.033). In contrast, the values of AFP declination showed no significant difference between groups (ΔAFP = 34.4 and 21.4 in complete and incomplete response group, respectively, *p* = 0.914).

### 3.2. Comparison of the HLs of DCP Associated with Standard Radiological Response

Because HLs could not be well assessed if pre-RFA DCP no more than two times the upper limit of normal (80 mAU/mL) or post-RFA DCP fell below to normal (40 mAU/mL) because HLs plateau reached after DCP normalization. Of the 109 patients, 42 were excluded due to pre-RFA DCP value of <80 mAU/mL, and 4 were excluded due to post-RFA value fell below 40 mAU/mL. The remaining 63 patients were evaluated to analyze the association between HLs of DCP and RFA efficacy.

The HLs of DCP in 63 patients’ *p* were calculated and were compared between groups according to radiological response at 1 and 12 months post ablation (Figure 2 and Figure 3). The ROC analysis of the HLs of DCP over complete radiological response was evaluated and showed that the optimal cut-off value of HLs for predicting complete radiological response was 47.5 h (area under the ROC curve: 0.865), with a sensitivity of 87.5% and a specificity of 72.1% (Figure 4). Therefore, we further defined short HLs of DCP as less than 48 h as a predictor of favorable RFA treatment response. Table 2 shows the difference in the HLs of DCP between complete and incomplete radiological response patients. Of the 43 RFA patients with a complete radiological response, 34 (79.1%) had short HLs of DCP < 48 h, and the remaining patients (20.9%) had long HLs of DCP ≥ 48 h. In 36 patients with short HLs of DCP, 34 (94.4%) had a complete radiologic response. However, among 27 patients with long HLs of DCP, only 9 patients had a complete radiologic response (33.3%). Based on the standard imaging modality, the sensitivity, specificity, accuracy, positive predictive value, and negative predictive value were 79.1%, 90.0%, 82.5%,94.4%, and 66.7%, respectively, for short HLs (<48 h) of DCP in the detection of a favorable RFA treatment response.

### 3.3. Analysis of the Discordant Results between Image Modality and the HLs of DCP

Eleven patients displayed discordant results, including two patients with incomplete radiological response with short HLs of DCP (<48 h) and the other nine with a complete radiological response with long HLs of DCP (≥48 h). Table 3 shows the series of DCP values of these patients (7 of 11 had four sets of DCPs at day 0, day 1, day 7 and day 30 after RFA, and four had three sets of DCPs). When analyzing the serial change in DCP levels, two patients with favorable short HLs of DCP and incomplete radiological response had early DCP levels re-elevated at day 7 or day 30. These results reminded the surgeon about the residual tumors even though they had favorable HLs of DCP on the first day post ablation but re-elevated DCP concentrations during follow-up. In addition, seven of nine (77.8%) patients who had complete response at 1 month post RFA but long HLs of DCP ≥ 48 h suffered from early tumor recurrence within 1 year post ablation. These results also suggested that we need to be aware of the early tumor recurrence in patients who had a complete radiological response at first but unfavorable DCP declination.

### 3.4. Association of Disease-Free Survival among the HLs of DCP

We performed a Kaplan–Meier estimation of long-term, recurrence-free survival. There were 14 (38.9%) and 24 patients (92.3%) who suffered from tumor recurrence; these patients had short HLs of DCP < 48 h and long HLs of DCP ≥ 48 h after 12 months post RFA, respectively. Patients who had short HLs of DCP < 48 h post ablation had a better disease-free survival curve than patients with long HLs of DCP ≥ 48 h (Figure 5) (*p*< 0.001, log-rank test).

### 3.5. Uni- and Multivariate Analysis of the Prognostic Factors for HCC Recurrence

Factors associated with recurrence-free survival are reported in Table 4. According to univariate analysis, median tumor size and shorter HL of DCP < 48 h were identified as prognostic factors for recurrence-free survival (*p* = 0.047 and 0.001, respectively). Among the tumor markers before and after RFA, neither AFP nor DCP levels showed significant difference in recurrence-free survival. In the multivariate analysis, shorter HL of DCP < 48 h (HR 0.12; 95% CI 0.03–0.44; *p* = 0.001) remained as independent prognostic factors of recurrence.

## 4. Discussion

RFA is an established, local curative treatment for the management of small or unresectable HCC [8,9]. However, the high incidence of tumor recurrence after RFA and unsatisfactory long-term survival are still problems for this percutaneous treatment [10,11]. Although the gold standard for the evaluation of treatment response is assessment by radiological imaging at 1 month post ablation using conventional criteria [35], there are limitations to using radiological evaluation to assess the treatment outcome, such as residual minute tumors in addition to the treatment site or microinvasion of cancer cells into vessels that could not be detected in a timely manner by currently available imaging techniques. Therefore, whether tumor markers could be predictors of treatment response and early recurrence continues to be an emerging issue in local therapies for HCC.

There are still conflicting data regarding whether pretreatment (such as hepatectomy or RFA) values of AFP or DCP could be a good prognostic marker to assess treatment response or recurrence-free survival in previous studies [36,37,38,39]. In one study by Lee et al. [40], the authors determined that only post-ablation DCP, rather than AFP, was an independent prognostic factor associated with overall survival. This result may reflect that serum AFP might increase after RFA due to hepatic necrosis caused by ablation.

Currently, we are more focused on the “change” in tumor marker concentrations between two time points (pre- and post-treatment) rather than their absolute levels, as a more useful marker for treatment response and early detection of recurrence. This is because it takes time for the values of tumor markers to decrease to the normal range after treatment, especially in patients with extremely high pretreatment levels, so the ratio of tumor marker concentration degradation may be better correlated to treatment response. In our previous study, we reported that a favorable AFP decrease reflects a good reference in RFA treatment response [29]. In Okamura and Tsukamoto’s studies, the authors calculated the HLs of DCP by pre- and post-resection DCP 1 month later and showed that prolonged HLs of DCP of more than 4 days was an independent prognostic risk factor for recurrence-free survival after HCC resection [41,42]. In our study, DCP and AFP levels displayed no significant difference between complete or incomplete RFA response group, neither at pre-RFA nor post-RFA. In contrast, the rate of DCP decrease was more significant in complete response group than that of the incomplete response group (Table 1, ΔDCP = 204.5 and 27.9 in complete and incomplete response group, respectively, *p* = 0.033). These results were comparable to previous reports that neither pre- nor post-treatment absolute DCP levels could reflect the treatment response or prognosis in resection for HCC.

The current study showed that short HLs of DCP (<48 h) are a good predictor for RFA response and early detection of residual tumor. In the present report, 34 of 36 patients with short HLs of DCP (94.4%) had a complete radiologic response. However, among 27 patients with long HLs of DCP, only 9 patients had a complete radiologic response (33.3%). This result suggested that short HLs of DCP may be a useful markers in the early detection of favorable RFA treatment response. During follow-up, patients with short HLs of DCP also had better positive prognostic significance for recurrence-free survival (RFS) after RFA therapies. In our study, the 12-month RFS was at 61.1% in short HLs group and 7.7% in long HLs group, which displayed significant difference in Kaplan–Meier curves analysis (*p* < 0.001) (Figure 5). The multivariate analysis also showed that short DCP half-life is a strong independent prognostic factor for recurrence-free survival (HR 0.12; 95% CI 0.03–0.44; *p* = 0.001) (Table 4). This result was consistent with previous studies showing that the time course of DCP changes had a significant correlation with tumor necrosis, treatment effect and prognosis [41,42,43].

To our knowledge, although many studies have reported that changes in DCP could predict the treatment response of HCC tumor resection or arterial infusion chemotherapy [41,42,44,45], the current study is the first report to focus on the association between HLs of DCP and RFA efficacy in HCC treatment. On the other hand, it was still a problem to determine appropriate checkpoint for HLs of DCP post treatment. In most previous studies, the post-treatment tumor markers were checked at 1 week to 4 weeks for evaluation of prognosis or treatment efficacy. Given that HL of DCP is short (2 to 4 days), the DCP levels possibly decline to a normal range within days after treatment, especially in patients with only two or three times the normal pretreatment DCP values. The present study is the first report on early check HLs of DCP on the day after RFA, and our results suggested that favorable HLs of DCP (<48 h) on the first day post ablation had the potential benefit of predicting treatment response and early recurrence in a timely manner. The utility of our findings in clinical use is to offer the advice that patients with long HLs of DCP after RFA might have higher risk of incomplete ablation and need further rescue therapy in time.

There are still some limitations of this study. First, because this was a single-center study and only patients with DCP ≥ 80 mAU/mL could be enrolled, the present report had limited cases and a less homogeneous patient population. Therefore, further multicenter studies with larger sample sizes are needed. Second, most HCC cases were not histologically confirmed in the current study. However, all HCC cases in this report were diagnosed following international guidelines. Third, the findings of the current study are not applicable to patients with normal or only mildly elevated pretreatment DCP levels. Therefore, whether the measurement of both DCP and AFP with HLs as predictors of RFA response has better sensitivity needs further study. Forth, in our study, we checked post-treatment DCP on the first day after RFA, which was evaluated at the earliest time point compared to other similar studies. Even though it remains a problem to determine the appropriate posttreatment checkpoint for DCP, our data showed that evaluating the HL of DCP at an earlier stage to detect incomplete treatment response of RFA in a timely manner had the potential benefit of predicting treatment response and early recurrence compared to AFP after ablation therapies.

## 5. Conclusions

In conclusion, the present study showed that short HLs of DCP (<48 h) are a useful predictive marker for treatment response and recurrence-free survival after RFA therapies. HCC patients who have long HLs calculated after RFA should be considered for further imaging modality studies to detect incomplete residual tumors and implement rescue management in a timely manner.

## Figures and Tables

**Figure 1 diagnostics-13-00696-f001:**
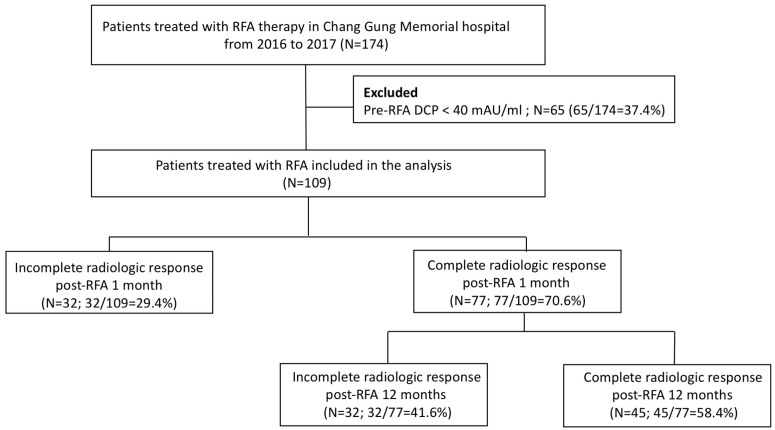
Flow diagram of enrolled patients with tumor marker Des-gamma-carboxy prothrombin (DCP) and radiofrequency ablation (RFA) therapies in this study.

**Figure 2 diagnostics-13-00696-f002:**
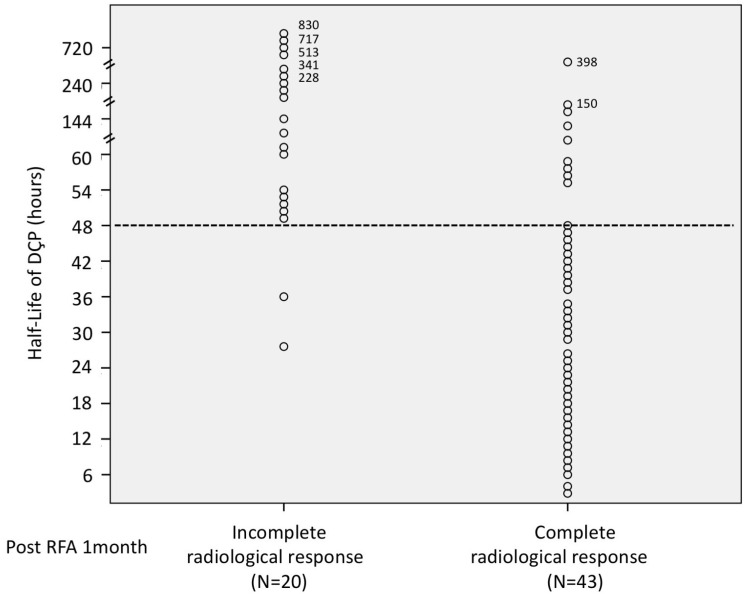
The distribution of 63 patients’ HLs of Des-gamma-carboxy prothrombin (DCP) stratified according to the radiofrequency ablation therapies (RFA) treatment response at 1 month post ablation.

**Figure 3 diagnostics-13-00696-f003:**
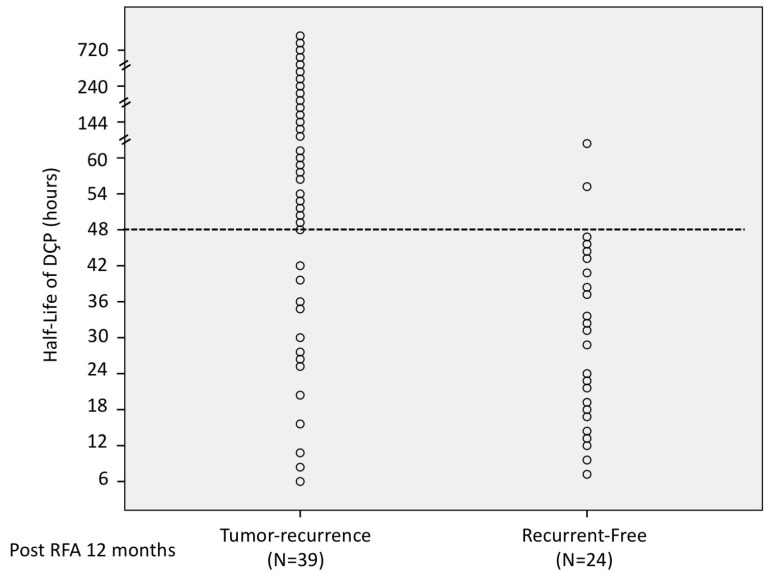
The distribution of 63 patient’s HLs of Des-gamma-carboxy prothrombin (DCP) stratified according to the tumor recurrence at 12 months post radiofrequency ablation therapy (RFA) ablation.

**Figure 4 diagnostics-13-00696-f004:**
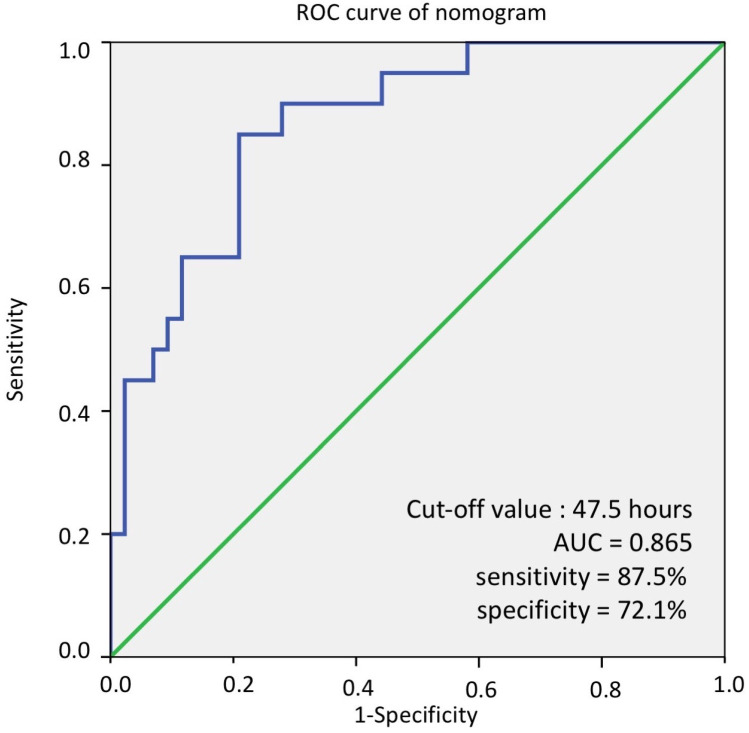
The ROC curve of nomogram for predicting response of RFA. The optimal cut-off value of HLs of DCP for predicting complete radiologic response of RFA was 47.5 h (AUC = 0.865), with a sensitivity of 87.5% and a specificity of 72.1%. ROC curve, receiver operating characteristic curve; RFA, radiofrequency ablation therapies; HL, half-life; DCP, Des-gamma-carboxy prothrombin; AUC, area under ROC curve.

**Figure 5 diagnostics-13-00696-f005:**
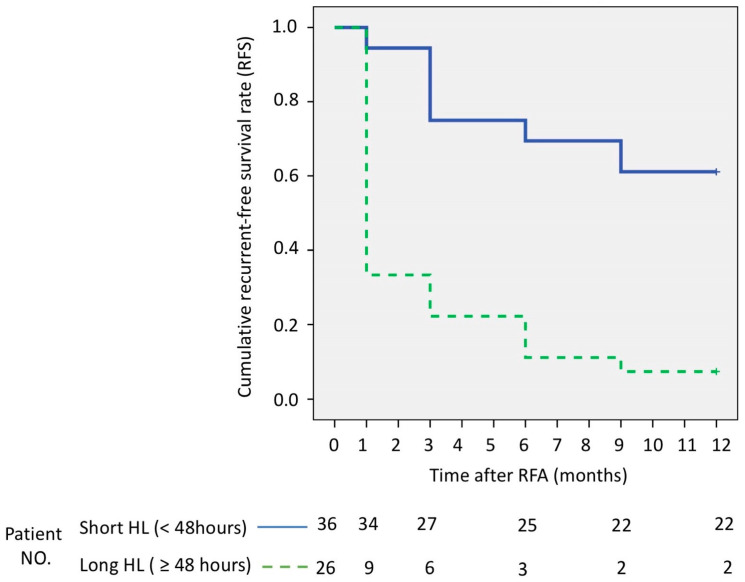
Comparisons of recurrent-free survival in 63 patients by half-life (HLs) of Des-gamma-carboxy prothrombin (DCP). Short HLs (<48 h) *n* = 36; Long HLs (≥48 h) *n* = 26.

**Table 1 diagnostics-13-00696-t001:** Baseline characteristics of 109 patients with radiofrequency ablation therapies (RFA) for hepatocellular carcinoma according to the radiological response at 1 month post RFA.

	RFA with Complete Radiological Response (N = 77)	RFA with Incomplete Radiological Response (N = 32)	*p*-Value
Age(years); mean ± SD	65.9 ± 9.4	69.1 ± 6.8	0.092
Sex(male); n(%)	49(63.6%)	24(75.0%)	0.251
BMI	24.6 ± 4.8	24.9 ± 3.3	0.799
Hypertension; n(%)	32(41.6%)	17(53.1%)	0.269
Diabetes mellitus; n(%)	25(32.5%)	10(31.3%)	0.901
Viral hepatitis; n(%)			0.365
HBs-Ag(−); HCV-Ab(−) HBs-Ag(+); HCV-Ab(−) HBs-Ag(−); HCV-Ab(+) HBs-Ag(+); HCV-Ab(+)	9(11.7%) 38(49.4%) 27(35.0%) 3(3.9%)	7(21.9%) 16(50.0%) 9(28.1%) 0 (0%)	
Cirrhosis; n (%) Non-cirrhotic Child A Child B ALBI grade; n(%) Grade 1 Grade 2	7(9.1%) 60(77.9%) 10(13.0%) 58(75.3%) 19 (24.7%)	2(6.2%) 26(81.3%) 4(12.5%) 16(50.0%) 16(50.0%)	0.879 0.012 *
Fibrosis-4(FIB-4) index	4.66 ± 4.26	6.36 ± 5.29	0.081
Ascites; n(%)	6(7.8%)	4(12.5%)	0.438
HCC naïve; n(%)	62(80.5%)	29(90.6%)	0.196
Max. tumor size (cm); mean ± SD	2.26 ± 0.82	2.39 ± 0.87	0.468
Max size > 2 cm; n(%)	43(55.8%)	18(56.3%)	0.969
Tumor Number; n(%) Solitary Multiple	54 (70.1%) 23 (29.9%)	15(46.9%) 17 (53.1%)	0.022 *
BCLC stage; n(%) Stage 0 Stage A Stage B	21(27.3%) 51(66.2%) 5(6.5%)	9(28.1%) 17(53.1%) 6(18.8%)	0.136
AJCC TNM stage; n(%) Stage Ia Stage Ib Stage II	25(32.5%) 35(45.5%) 17(22.0%)	10(31.2%) 7(21.9%) 15(46.9%)	0.018 *
AST(IU/L)	42.2 ± 25.4	49.3 ± 34.9	0.242
ALT (IU/L)	37.9 ± 30.1	39.9 ± 31.1	0.752
Total bilirubin (mg/dL)	1.01 ± 0.74	1.16 ± 0.76	0.318
Hemoglobin (gm/dL)	12.8 ± 2.3	12.5 ± 2.1	0.549
Platelets (×10^9^/L); mean ± SD	146.9 ± 82.7	117.7 ± 62.4	0.076
Prothrombin time (sec); mean ± SD	10.7 ± 0.65	10.8 ± 0.91	0.303
Albumin (g/dL)	4.19 ± 0.53	4.04 ± 0.51	0.193
Pre-RFA AFP > 400 ng/mL; (n%)	9(11.7%)	5(15.6%)	0.576
Pre-RFA AFP (ng/mL); mean ± SD	271.3 ± 885.4	460.4 ± 663.6	0.293
Post-RFA AFP (ng/mL); mean ± SD	231.3 ± 803.9	441.5 ± 349.0	0.317
Value of AFP declination; Δ mean± SD	Δ 34.4 ± 73.9	Δ 21.4 ± 93.3	0.914
Pre-RFA DCP(mAU/mL); mean ± SD	639.4 ± 2210.5	512.9 ± 1340.2	0.764
Post-RFA DCP(mAU/mL); mean ± SD	437.3 ± 1463.3	496.5 ± 1304.3	0.843
Value of DCP declination; Δ mean ± SD	Δ 204.5 ± 770.9	Δ 27.9 ± 55.7	0.033 *

Abbreviations: BMI, body mass index; HBs Ag, hepatitis B virus antigen; HCV-Ab, hepatitis C virus antibody; ALBI grade, albumin–bilirubin grade; HCC, hepatocellular carcinoma; SD, standard deviation; BCLC stage, Barcelona clinic liver cancer classification; AJCC, American Joint Committee on Cancer. AST, aspartate aminotransferease; ALT, alanine aminotransferase; RFA, radiofrequency ablation therapies; AFP, alpha-fetoprotein; DCP, Des-gamma-carboxy prothrombin. * means statistically significant difference, *p* < 0.05.

**Table 2 diagnostics-13-00696-t002:** Difference in DCP half-life between complete and incomplete response RFA patients by standard image modality.

DCP Half Life (HL)	Complete Radiological Response	
	Yes (N = 43)	No (N = 20)
Shorter HL < 48 h	34	2
Longer HL ≥ 48 h	9	18

Abbreviations: DCP, Des-gamma-carboxy prothrombin; RFA, radiofrequency ablation therapies; HL, half-life.

**Table 3 diagnostics-13-00696-t003:** Series of DCP values of 11 patients with discordant results between HLs of DCP and RFA efficacy.

Case	Pre-RFA (D0)	Post-RFA (D1)	Post-RFA (D7)	Post-RFA (D30)	HL of DCP (Hours)	Recurrent-Free Survival (Months)
Patients with shorter HL of DCP (<48 h) but incomplete RFA response post RFA, 1 month						
Case A	177.85	121.24	151.78	355.83	27.28	1
Case B	349.60	264.30	Lost	303.91	36.13	1
Patients with longer HL of DCP (≥48 h) but complete RFA response post RFA, 1 month						
Case 1	397.42	326.73	171.16	114.74	54.6	12
Case 2	914.76	858.79	Lost	429.59	152.1	6
Case 3	140.61	130.60	90.81	148.92	126.3	6
Case 4	356.12	318.94	351.54	Lost	83.9	3
Case 5	2181.46	1986.12	984.89	817.17	90.8	6
Case 6	150.51	140.99	55.22	32.77	114.8	3
Case 7	183.99	152.19	104.99	76.85	53.9	12
Case 8	162.31	144.10	Lost	66.76	91.6	9
Case 9	89.91	83.08	Lost	118.40	150.1	3

Recurrent-Free survival (RFS) was defined as the time interval from RFA treatment to first recurrence. Patients were followed up every 3 months by standard modality for 12 months. Abbreviations: DCP, Des-gamma-carboxy prothrombin; RFA, radiofrequency ablation therapies; HL, half-life.

**Table 4 diagnostics-13-00696-t004:** Univariate and multivariate analysis of prognostic factors for recurrence-free survival.

Variables	Univariate Analysis		Multivariate Analysis	
	HR (95% CI)	*p* Value	HR (95% CI)	*p* Value
Age (yrats)	1.01(0.94–1.07)	0.845		
Sex (male)	0.39(0.14–1.14)	0.084		
Cirrhosis (present)	1.62(0.29–9.08)	0.585		
ALBI grade	0.65(0.25–1.66)	0.367		
Fibrosis-4 index	0.99(0.89–1.10)	0.854		
Ascites (present)	2.64(0.41–17.1)	0.308		
AST (IU/L)	0.98(0.96–1.02)	0.336		
Alt (IU/L)	1.02(0.98–1.05)	0.255		
Albumin (g/dL)	1.08(0.39–2.95)	0.876		
Total bilirubin (mg/dL)	0.43(0.18–1.04)	0.062		
Platelet counts (×10^9^/L)	1.00(0.99–1.01)	0.903		
Prothrombin time (seconds)	0.96(0.49–1.87)	0.902		
Max tumor size > 2 cm	0.70(0.25–2.00)	0.505		
Median tumor size (cm)	3.69(1.02–13.4)	0.047		
Tumor Number (multiple)	0.58(0.20–1.68)	0.317		
BCLC stage	1.07(0.54–2.12)	0.854		
Pre-RFA AFP (ng/mL)	0.98(0.99–1.02)	0.631		
Pre-RFA DCP (mAU/mL)	1.01(0.99–1.02)	0.072		
Post-RFA AFP (ng/mL)	0.99(0.98–1.01)	0.630		
Post-RFA DCP (mAU/mL)	0.98(0.97–1.01)	0.073		
Short HL of DCP (<48 h)	0.13(0.04–0.43)	0.001	0.12(0.03–0.44)	0.001

Abbreviations: ALBI grade, albumin–bilirubin grade; AST, aspartate aminotransferease; ALT, alanine aminotransferase; BCLC stage, Barcelona clinic liver cancer classification; RFA, radiofrequency ablation therapies; AFP, Alpha-fetoprotein; DCP, Des-gamma-carboxy prothrombin.

## Data Availability

The datasets analyzed in current study are available from the corresponding author on reasonable request.

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
