# Peer review of "Short Half-Life of Des-γ-Carboxy Prothrombin Is a Superior Factor for Early Prediction of Outcomes of Hepatocellular Carcinoma Treated with Radiofrequency Ablation"

_diagnostics, 2023, doi:10.3390/diagnostics13040696_

Round 1

Reviewer 1 Report

I read with attention Yao et al's work.

It was interesting, authrs surrounded the problem, the limitations of their work. Despite these difficulties, these results need to be known and discussed.

Some minor revision can be proposed:

- Introduction is too long. authors can be more synthetic and some data would be preferentially used in discussion.

-Delta-AFP would be probably not sufficient, but it must be mentioned in the table.

Author Response

Thank you for offering the opportunity to revise our manuscript.We have revised it according to the comments.We look forward to hearing from you if there is any further query.

Best regards,

Point 1: Introduction is too long. authors can be more synthetic and some data would be preferentially used in discussion.

 Response 1:

Thanks very much. We revised the introduction part of our manuscripts and produsing our report that are more synthetic ,and concise . Besides, we also re-edited some data and reference from introduction to discussion section.

The revised part was be adjusted in the revised manuscript: “Introduction” section , “discussion” section, and “reference” section.

Point 2: Delta-AFP would be probably not sufficient, but it must be mentioned in the table.

Response 2:

Thanks very much. We had supplemented the data of value of AFP declination and the statitstical result. The value of AFP declination were 34.4 and 21.4 in complete and incomplete RFA group,respectively. P=0.914.

The revised part was be supplemented in the revised manuscript: “Result” section , and “Table 1” section.

Reviewer 2 Report

Interesting manuscript in which the authors show that short half-life of des-γ-carboxy prothrombin -DCP- is effective in early prediction of complete radiological response and disease-free survival in patients with hepatocellular carcinoma treated with radiofrequency ablation.

1.      The text is written correctly and clearly, the rationale of the study is explained exhaustively, the materials and methods are explained in detail.

2.      In the results, the analysis of the impact of DCP variations on disease free survival was performed only with Kaplan Meier, while a multivariate analysis that would be useful is missing.

3.      Furthermore, it would be interesting to know the impact of the radiological response on disease-free. Indeed, what is not clear is the usefulness of what the authors discovered. Predicting radiological response is of questionable value, since results in terms of response are available in a short time anyway.

4.      Instead, for disease-free prediction, it would be useful to know whether changes in DCP add predictive power to traditional prognostic factors and the impact, if any, of clinical response on disease-free.

5.      More generally, it would be helpful if the authors explained the utility of their findings in terms of their use in clinical practice, also considering the fact that patients with initial low levels of DCP were excluded from the analysis.

6.      Finally, I would urge authors to double-check tables and figures, since there is usually a lack of legends that explain abbreviations.

Author Response

Thank you for offering the opportunity to revise our manuscript.We have revised it according to the comments.We look forward to hearing from you if there is any further query.

Best regards,

Point 1: The text is written correctly and clearly, the rationale of the study is explained exhaustively, the materials and methods are explained in detail.

Response 1:

Thank you. I appreciate the compliment.

Point 2:  In the results, the analysis of the impact of DCP variations on disease free survival was performed only with Kaplan Meier, while a multivariate analysis that would be useful is missing.

Response 2:

Thanks very much. We had revised the manuscript and supplemented the data

of uni and multivariate analysis of prognosic factors for recurrence-free survival.

The multivariate analysis showed that short DCP half-life is a strong independent prognostic factors for recurrence-free survival (HR  0.12; 95% CI 0.03-0.44 ; P = 0.001)

 The revised part was be supplemented in the revised manuscript: “Result” section , “discussion” section  and “Table 4” .

Point 3: Furthermore, it would be interesting to know the impact of the radiological response on disease-free. Indeed, what is not clear is the usefulness of what the authors discovered. Predicting radiological response is of questionable value, since results in terms of response are available in a short time anyway.

Point 4: Instead, for disease-free prediction, it would be useful to know whether changes in DCP add predictive power to traditional prognostic factors and the impact, if any, of clinical response on disease-free. 

Response 3 and 4:

Thanks very much, and I agree with your comment about the usefullness of predicting radiological response post ablation therapy. Indeed, most paper discussed about the useful factors for predicting tumor prognosis or recurrence-free surivial. Our paper not only found that short DCP half-life is consistent with complete radiological response , but also a strong prognostic factors for recurrence-free survival. A new point in our report is that we took advantage of short HL of DCP (48-72 hours) and tried to figure out the appropriate/earliest check-point for tumor makers follow up post-treatment. The present study is the first report of early check HLs of DCP on the day after RFA, and our results suggested that favorable HLs of DCP (< 48 hours) on the first day post-ablation had the potential benefit of predicting treatment response. The usefulness of this new insight is that we could detected possible incomplete RFA early by short HL of DCP, then might used other diagnostic tool (like contrast enhenced ultrasound) to confirm residual tumor ,and give patient further treatment in time.

Point 5: More generally, it would be helpful if the authors explained the utility of their findings in terms of their use in clinical practice, also considering the fact that patients with initial low levels of DCP were excluded from the analysis.

Response 5:

 Thanks very much. We revised the manuscript of discussion part and supplemented our idea of utility of shoft HL of DCP and their usefulness in clinical practise.The revised part was be supplemented in the revised manuscript: “discussion” section.

Besides,about the exclusion criteria, half-life could not be well assessed by formula if pre-RFA DCP no more than 2 times the upper limit of normal (80 mAU/ml) or post-RFA DCP fell below to normal (40 mAU/ml) because HLs plateau reached after DCP normalization, so we excluded DCP which levels < 80 mAU/ml or post-RFA DCP fell below to normal (40 mAU/ml) in further analysis. In manuscript: “Result” section : comparision of HLs of DCP with standard radiological response.

Point 6 :Finally, I would urge authors to double-check tables and figures, since there is usually a lack of legends that explain abbreviations.

Response 6:

Thanks very much.We had revised our tables and figures and supplemented the abbreviations. 

Round 2

Reviewer 2 Report

The quality of the manuscript improved after review by the authors.